# Risk of COVID-19 in Chagas Disease Patients: What Happen with Cardiac Affectations?

**DOI:** 10.3390/biology10050411

**Published:** 2021-05-06

**Authors:** Alejandro Diaz-Hernandez, Maria Cristina Gonzalez-Vazquez, Minerva Arce-Fonseca, Olivia Rodriguez-Morales, Maria Lilia Cedilllo-Ramirez, Alejandro Carabarin-Lima

**Affiliations:** 1Centro de Investigaciones en Ciencias Microbiológicas, Instituto de Ciencias, Benemérita Universidad Autónoma de Puebla, 14 Sur y Avenida San Claudio, Ciudad Universitaria, Puebla 72570, Mexico; alexhalo6@hotmail.com (A.D.-H.); crispi333@yahoo.com.mx (M.C.G.-V.); lilia.cedillo@correo.buap.mx (M.L.C.-R.); 2Departamento de Biología Molecular, Instituto Nacional de Cardiología “Ignacio Chávez”, Juan Badiano No. 1, Col. Sección XVI, Tlalpan, México City 14080, Mexico; mini_arce@yahoo.com.mx (M.A.-F.); rm.olivia@gmail.com (O.R.-M.)

**Keywords:** Chagas disease, COVID-19, coinfections, cardiopathies, cardiac complications

## Abstract

**Simple Summary:**

The SARS-Cov-2 virus appeared as a pandemic at the end of 2019. Since then, the literature on the effects of this disease has been increasing. Chagas disease is more than 110 years old since its discovery, and the implications of these patients when co-infected with SARS-Cov-2 have not been reported. In this review, we summarize the studies to date on the cardiac affectations generated by both diseases in humans, describing their possible interrelation, the damage that coinfection could generate, the analysis of the treatment for both diseases and recommendations to avoid SARS-Cov-2 infection in patients with Chagas disease. This is the first comprehensive review of cardiac disorders that can manifest in chagasic patients as a result of coinfection.

**Abstract:**

Background: Chagas disease is considered a neglected tropical disease. The acute phase of Chagas disease is characterized by several symptoms: fever, fatigue, body aches, headache and cardiopathy’s. Chronic phase could be asymptomatic or symptomatic with cardiac compromise. Since the emergence of the pandemic caused by the SARS-CoV-2 virus, the cardiovascular involvement has been identified as a complication commonly reported in coronavirus disease 2019 (COVID-19). Due to the lack of knowledge of the cardiac affectations that this virus could cause in patients with Chagas disease, the aim of this review is to describe the possible cardiac affectations, as well as the treatment and recommendations that patients with both infections should carry out. Methods: The authors revised the recent and relevant literature concerning the topic and discussed advances and limitations of studies on COVID-19 and their impact in Chagas disease patients, principally with cardiac affectations. Results: There currently exists little information about the consequences that Chagas disease patients can suffer when they are infected with COVID-19. Conclusions: This review highlights the emerging challenges of access to medical care and future research needs in order to understand the implications that co-infections (SARS-CoV-2 or other viruses) can generate in Chagas disease-infected people.

## 1. Introduction

Latin America is an endemic region for Chagas disease [1], which is caused by the protozoan *Trypanosoma cruzi* and represents a complex public health problem due to the lack of vaccines and effective treatments. *T. cruzi* has the ability to affect multiple body organs, such as cardiovascular and digestive systems [2] mainly in adult populations, as well as people with little access to health and a high rate of comorbidities. These effects are very similar for the development of coronavirus disease 2019 (COVID-19), which is caused by Severe Acute Respiratory Syndrome Coronavirus-2 (SARS CoV-2) and, since 2019, has spread throughout the world at an alarming rate, bringing many social, economic, epidemiological and health changes.

The first reported case of COVID-19 in Latin America was in São Paulo, Brazil on 25 February 2020 [3]. In Mexico City, it occurred two days later on 27 February [4]. By 2 April 2021, 130,108,655 positive cases and 2,836,363 global deaths had been reported; meanwhile, for Mexico, 2,244,268 positives cases and 203,664 deaths had been reported [5]. This increase in the rate of infection never seen before puts the underestimated population of patients with Chagas disease in Mexico at a clear disadvantage, because it increases considerably the rate of morbidity and mortality. Chagas disease is divided into two phases, acute and chronic, the latter consisting of an asymptomatic stage (formerly called indeterminate) and the symptomatic stage [6] represent an important deterioration to health, which makes the patient more susceptible to infection by the SARS-CoV-2, especially in chronic indeterminate phase due to the patient does not know they have *T. cruzi* infection.

Chagas disease belongs to the group of Neglected Tropical Diseases (NTDs), which tend to disproportionately affect sectors of the population with poor or null access to health, having an impact on early childhood development, pregnancy and worker productivity [7]. In 2018, it was estimated that 52.4 and 9.3 million Mexican people live in moderate and extreme poverty, respectively, and 20.2 million have limited or null access to health services in the country. The Mexican states with the highest rates of moderate poverty are Chiapas, Guerrero, Oaxaca and Veracruz, with 76.4%, 66.5%, 66.4% and 61.4%, respectively [8]. These states represent key regions where there is a greater registry of the vector of Chagas disease, hematophagous bugs belonging to the subfamily Triatominae. Up to 31 vector species are known in Mexico, 19 of epidemiological importance and 13 directly related to *T. cruzi* infections [9].

In Mexico, it is estimated that there are 1.1 to 5.5 million people infected with *T. cruzi* [2,10,11]. Up to the 53rd epidemiological week of 2020, 391 positive cases of Chagas have been registered in Mexico (both acute and chronic forms) [12], and it is estimated that, throughout the country, 30 million people are at risk of contracting the disease due to the fact that Chagas disease is endemic in our country [9].

This combination of poverty, endemicity, vector-prone regions and vulnerable population is the main reason why we analyzed the impact of the COVID-19 pandemic on the Mexican population suffering from Chagas disease.

The objective was to review the main condition of Chagas disease such as Chronic Chagas Cardiomyopathy (CCC) and its relationship with the pathologies caused by COVID-19, as well as the influence of treatment against COVID-19 in patients with Chagas disease, and some recommendations both to prevent the infection and to contend with both diseases.

## 2. Phases of the Chagas Disease

Chagas disease has a great capacity to spread being transmitted by different routes, either through blood-feeding insects, blood transfusions, organ donation, orally through food and even accidentally with contaminated samples [7]. On the other hand, the main Discrete Typing Unit (DTU) (molecular genotyping that allows to identify particular characteristics of the isolated trypanosomes) of the parasite found in Mexico is the *T. cruzi* I (TcI) [13], which has a marked tropism towards the heart [14], making this organ one of the main anatomical sites that is most affected by the protozoan infection.

Once the parasite is inoculated, the acute phase of the disease occurs, which usually takes 15–40 days; most of these cases are asymptomatic and occur during childhood [15,16]. During this phase, we can observe or not the following signs and symptoms: Inflammation around the inoculation site (chagoma), unilateral palpebral edema (Romaña sign), fever, headache, joint and muscle pain, anorexia, vomiting, diarrhea, drowsiness, apathy, lymphadenopathy, hepatosplenomegaly, edema and convulsions [17,18]. This whole range of symptoms makes Chagas disease very difficult to diagnose, except for Chagoma and Romaña sign, which are the disease pathognomonic signs. In this phase, deaths usually occur in 5–10% of infected people, generally from severe myocarditis, meningoencephalitis or both [19].

About 60–70% of those infected do not develop signs or symptoms, which is why they pass directly the indetermined chronic phase. This phase is characterized by a seropositivity of antibodies against *T. cruzi*; a normal 12-lead electrocardiogram and a normal radiology examination of the chest, esophagus and colon [19]. The period of duration for the chronic indeterminate phase can vary from 10–30 years, and even the patient can die derived from other complications and never know that he has had Chagas disease. This is a rather critical phase for the population; since the patient is not aware of the risk that their illness represents, ignorance about their condition also represents a major challenge for the health system: to detect the infection as early as possible. It is believed that, in Mexico, there has been considerable transmission of Chagas disease through blood transfusion and by organ donation due to the lack of regulation for the detection of *T. cruzi* for many years. It was not until the norm NOM-253-SSA1-2012, which dictates mandatory screening of blood donors, was applied that a greater number of cases were detected, and infections were avoided through this route [9].

The remaining 30–40% of those infected with *T. cruzi* will go into the symptomatic chronic phase. Although this is usually the progression followed by a Chagas patient, direct progressions from the acute phase to clinical forms, i.e., cardiomyopathy, megaesophagus, megacolon and cardiodigestive disorders, have been detected in only 5–10% of patients. There is a different epidemiological distribution in the American continent that also makes a difference in the clinical presentations of Chagas disease. The digestive affectations are mainly, but not exclusively, distributed in the South America region (mainly in Argentina, Brazil and Bolivia), and they are found in 10–15% of patients infected in a chronic way. For the pathologies found in Central and North America, there are mainly cardiopathies which develop in 20–30% of the patients, being the most common disorder in Chagas disease [2,6,19]. This variation in pathologies can be explained by a tropism defined by the *T. cruzi* strain, of which the full mechanism is still being elucidated [14].

### Acute and Chronic Chagas Cardiomyopathy

It is well-known that the heart is the most frequently affected organ during chronic Chagas disease [20], although cardiac affectations can also be detected in the acute phase. The complexity of the *T. cruzi* life cycle, which involves four distinct morphologically and biochemically well-characterized forms [20], is reflected in the complexity of the infection, which is a pathological condition with many variables including both the parasite and the host and its immune system.

In the acute phase of heart Chagas disease of mammals, the process initiates with the inoculation of the parasite. *T. cruzi* metacyclic trypomastigotes located in the perianal region of the triatomine vector are released in the feces or urine of the vector after feeding on the host’s blood. These forms can infect the mammalian host through two routes, by contact with mucosa or by penetrating through localized wounds in the epithelium. Once inside the host, the parasite rapidly invades a wide variety of nucleated mammalian cells, including myocytes, endothelial cells, neurons, fibroblasts and adipocytes. These cells are the ones that reside or are recruited into that tissue next to the wound skin [11,19] (Figure 1).

As previously mentioned, the most common DTU of *T. cruzi* found in Mexico has a marked tropism towards cardiac muscle, specifically to striated cardiac myofibrils. Despite its very rare occurrence (<1%), apparent myocarditis developed during the acute phase has been observed [21], in which myocardial inflammation is similar to those presented by other forms of myocarditis, including those caused by other infectious agents, cardiotoxins and radiation [22]. Once the parasite has infected heart tissue, a fulminant intracardiac replication can occur, with high numbers of amastigote forming amastigote nests, which may lead to myonecrosis, myocytolysis, myofibrils degeneration and intense vasculitis. This will depend largely on the genetic and immunological variants of the parasite and on the host, whose defense system could trigger an intense inflammatory response, which consists primarily of leukocytes, including eosinophils, lymphocytes, macrophages and mast cells, accompanied by increased expression of inflammatory mediators such as cytokines, chemokines and nitric oxide synthase that also complement activation, antibody production and opsonization [20,22,23]. As the disease continues its course, leaving the acute phase and also giving rise to the adaptive immunity, the number of parasitized cells and the inflammatory response decrease considerably, bringing the myocardial tissue to a near-normal state and leaving possible abnormalities such as scarring (interstitial fibrosis), myofiber hypertrophy and continuous minimal inflammation [21]; the whole process usually takes two to four months [20]. In general terms, the characteristics of the acute-phase Chagas cardiomyopathy can range from minimal changes in the heart muscle to dilation of the atria and ventricles in the most severe cases (cardiomegaly); there may also be bundle branch block, diffuse changes of segment ST and T wave, prolonged PR interval and sinus tachycardia [22].

The next step in the course of the disease is chronicity, where the CCC is developed. While this review focuses mainly on heart disorders, it is important to remember that there is another relevant chronic disorder, digestive, which is not so common in Mexico. In chronic phase, many people can live their whole life without any symptoms, most of them without knowing about their infection that probably happened in their childhood.

Once the infection has been present for a few decades, a slow process of inflammation and scarring of the myocardium begins, leading to general changes of heart morphology in approximately 20–30% of infected individuals [21]. These patients may or may not present ventricular failure, a process that is currently classified into five stages according to international recommendations based on ventricular dysfunction [22]. In stage A, which is the only within the chronic indeterminate form, positive serology for Chagas is detected, but no structural heart disease, neither symptoms of heart failure, the electrocardiogram and X-ray are both normal. The stage B includes patients with structural heart disease but who have never had signs or symptoms of heart failure; this phase is divided into two: B1, patients with abnormalities in the electrocardiogram or echocardiography but still without ventricular dysfunction or heart failure, and B2, also with structural heart disease, without signs or symptoms of heart failure, but with a global ventricular dysfunction. Stage C is characterized by ventricular dysfunction and signs or symptoms of heart failure. Finally, stage D encompasses advanced heart failure at rest, despite optimal medical treatment and will therefore require specialized interventions [17,22]. Hearts of patients with chronic Chagas disease are usually fully enlarged, with a dilation of the chamber that predominates over hypertrophy, giving a globular shape; in approximately half of the people who develop chronic disease, the heart shows an unusual thinning of the ventricular wall, which protrudes in the form of an aneurysm [20]. The most frequent and general pathological features of the CCC include low-grade myocarditis associated with myocytolysis, myofiber hypertrophy and interstitial fibrosis [18]. In contrast to the acute phase, in the chronic phase usually only focal areas of inflammation are found that are mainly composed of T cells and macrophages, with a few eosinophils, plasma cells and mast cells; moreover, parasites are rarely observed by conventional microscopic analysis of cardiac biopsy samples [21]. In the final stage of this phase, after suffering severe damage, some patients may need a heart transplantation. The first transplant to end-stage CCC patient was performed in 1985 at the Heart Institute in São Paulo, Brazil [24]. Although initially, it seems that a transplant would resolve the cardiac damage by *T. cruzi*, the risk of reactivation is quite high and common. The reactivation phenomenon, reported in the early 1960s, occurs mainly in patients who are under cellular suppression, either by whole-body irradiation, hematological malignancies, chemotherapy and immunologically compromised, such as those coinfected with the Human Immunodeficiency Virus (HIV) [24]. The most frequent clinical manifestations in Chagas disease reactivation are usually meningitis, encephalitis [24], myocarditis, inflammatory panniculitis, skin nodules and even fever, signs and symptoms that lead to allograft dysfunction and probably to a rapid onset of heart failure [25]. Reported rates of reactivation in heart transplant recipients with chronic Chagas disease in Latin America vary widely from 20–90%, and the time from transplantation until reactivation of Chagas disease ranges from 11 to 23 weeks [25,26].

According to the most conservative estimations, in Mexico, there are 1.1 million people infected by *T. cruzi*, both in the acute and chronic phases. Since 2017, the Mexican government has registered both acute and chronic cases in SINAVE Epidemiological Bulletin [12]. However, these reported cases are below the predicted estimates. On the other hand, there is no correlation of heart diseases associated with Chagas disease in its acute and chronic forms, so the estimates indicate that a high percentage of these heart diseases have an origin associated with Chagas disease (Figure 2), and although there are no official data in Mexico, presumptively, some authors have predicted the percentage of cardiac affectations due to Chagas disease, with an approximate of <1% of cardiomyopathies derived from infection in the acute phase [21] and 25% ± 5% in the chronic phase [19]. It is important to consider that the asymptomatic chronic phase contributes with more cases that are not identified or recorded, and therefore, it is possible that the total data of *T. cruzi* infections are underestimated.

The figure represents the total reported *T. cruzi* infections in both acute and chronic phases; these data are based on the epidemiological bulletins published by the Ministry of Health in Mexico [12], as well as estimated cardiopathies related to Chagas disease that were predicted according to that described by Rassi et al. 2010 [19] and Bonney et al. 2019 [21], who propose that the prevalence of cardiopathies associated with Chagas disease corresponds to 25% in the chronic phase and less than 1% in the acute phase of the total of reported cases, respectively. The estimated cases of cardiopathies in the chronic phase are represented by double asterisks and with an asterisk the estimated cases of cardiopathies in the acute phase of Chagas disease.

## 3. Cardiac Implications, Patients with COVID-19 and Chagas Disease

Cardiac implications in Chagas disease were explained in previous sections, but as far as COVID-19 is concerned, the information is still in its infancy. One of the most studied processes of SARS CoV-2 infection so far is the binding to the human receptor angiotensin-converting enzyme 2 (ACE2), which is mainly expressed in the lungs, vascular endothelium and heart [27].

The general infection process starts primarily with the spike (S) protein protruding from the viral surface, and it is composed of homotrimers consisting of two subunits S1/S2, where S1 contains the receptor-binding domain (RBD), and it is responsible for binding to host cell surface receptors, mainly ACE2 [28,29]. After this binding, the S protein is cleaved at its polybasic site, mainly by the cell surface-associated transmembrane protease serine 2 (TMPRSS2) but, also, through the cellular cathepsin L, thus activating the S2 subunit, which mediates the fusion between the viral and host cellular membranes. Cathepsin L then activates the remaining S protein into the endosomes and can compensate the entry in cells lacking TMPRSS2 [29,30]. Once the genome is released into the cytosol, open reading frames 1a and 1b (ORF1a and ORF1b) are translated into viral replicase proteins, which are cleaved into nonstructural proteins by the action of host and viral papain-like proteases; these proteins constitute RNA-dependent RNA polymerase. Then, the replicase components rearrange the endoplasmic reticulum into double-membrane vesicles that facilitate viral replication of genomic and sub-genomic RNAs; the latter are translated into accessory and viral structural proteins that facilitate virus particle formation. Finally, subsequent positive-sense RNA genomes are incorporated into new synthesized virions that are secreted from the plasma membrane [29,31,32].

The general symptoms of COVID-19 are similar to those caused by SARS-CoV [33] and the Middle East Respiratory Syndrome Coronavirus (MERS-CoV) [28,34]; however, SARS CoV-2 infection has a relatively lower fatality rate with ~2%; about 20% of patients develop multiple severe conditions such as acute respiratory distress syndrome and multiple organ failure [35], and 23% of patients in serious conditions have heart damage [36]. Although this is current information and is constantly updated based on studies and scientific contributions, as well as new mutants of the virus that may or may not have a significant impact on its pathogenesis, it is very likely that the main damage to heart tissue caused by SARS CoV-2 infection is based on binding to the ACE2 receptor found in type 2 pneumocytes, macrophages and cardiomyocytes, as well as perivascular pericytes [37] that seem to express more of this receptor; these cells are located outside the endothelial wall of the capillary and part of the venules. Pericytes can play an essential role in myocardial microcirculation [34], since the binding of the virus to the ACE2 receptor could trigger an inflammatory response that, in turn, can lead to myocardial dysfunction and damage, dysfunction endothelial disease, microvascular dysfunction, plaque instability and myocardial infarction [38]. Thus, the immune status caused by COVID-19 could act as a possible trigger for the progression of Chagas disease, perhaps influenced by certain parasitic factors, such as the type of strains, the parasite load and the presence of the parasite circulating in the bloodstream, as well as host factors, such as genetic susceptibility and immune status, specifically gamma-IFN [37]. There is also the possibility that even though parasitemia is absent (i.e., the chronic form), it could be reactivated due to pharmacological and disease-induced immunosuppression [39]. This potential reactivation could be caused by a severe disease similar to hemophagocytic lymphohistiocytosis (cytokine storm) during the initial immune response [40], in which there is cytokines and chemokines (IL-6, TNF-alpha and CXCL 10) production frequently related to the inflammatory immune response stimulated in CCC pathogenesis [41,42]. This condition can also be due to the presence of some viruses [43] or even the use of some COVID-19 treatments such as steroids [44], hydroxychloroquine and other immunomodulatory drugs, mainly interleukins inhibitors, which have been associated with the progression of Chagas disease [37,45]. Another important issue is the manifestation of arrhythmias, increasingly detected in COVID-19 patients [38], probably caused by proinflammatory cytokines and T-lymphocyte cardiotropism, believed to arise from the interaction between heart-produced hepatocyte growth factor and the c-Met, a receptor on naive T lymphocytes [46]. Although the pathophysiology of these arrhythmias is not entirely clear, if left unchecked, they can lead to fulminant heart failure and cardiogenic shock, which is thought to be dependent of the stage of myocardial injury and the production of proinflammatory cytokines [47] both conditions present in the CCC.

Another important aspect to consider during cardiopathies and their severity in Chagas disease is the levels of the ACE2 receptor, since some studies directly correlate with them [48,49]. Injuries such as heart failure are characterized by the activation of peptide systems, such as the renin-angiotensin system (RAS) and, particularly, the B-type natriuretic peptide (BNP) systems, as well as atrial natriuretic peptide (ANP) [48]. BNP and ANP act as potent molecular markers in patients with Chagas disease and other types of dilated cardiomyopathies. RAS acts as a master regulator for blood pressure, volume and electrolyte homeostasis. One of the components of this system is the ACE2 receptor, which is not only expressed in organs but, also, in blood vessels [48,50]; some studies have demonstrated the solubility of ACE2 in plasma, showing a significant increase in patients with heart failure and a correlation with the degree of severity of the disease [51], giving better results as a biomarker in the diagnosis and prognosis of patients with Chagas disease by independent or combined detection with BNP or ANP. Studies in chagasic patients with systolic dysfunction have shown a slight increase in ACE2 levels compared to healthy individuals; however, these levels were significantly increased in sera from patients who already had more serious clinical manifestations of heart failure, giving the ACE2 receptor an important predictive value for the necessity of cardiac transplantation and mortality rate in patients with chronic Chagas disease [48,49]. This correlation between the severity of cardiac damage and the increase in the level of ACE2 can be explained by the regulatory role of this receptor in cardiac function and remodeling, both being commonly affected by the presence of the parasite. Additionally, although ACE2 is usually an integral membrane protein, some studies show that it can be cleaved by a protease to be released into the blood serum [52]. Other studies show that blocking the production or inhibiting the effects of Angiotensin II (Ang II) could block or delay cardiac remodeling, reducing mortality and morbidity, as demonstrated experimentally by the exogenous addition of the vasodilator Ang (1–7), which improved cardiac function after myocardial infarction [49,53]. Some beta-blockers, such as Carvedilol and Metoprolol, have been shown to inhibit the RAS, thereby reducing natriuretic peptides and improving systolic ventricular function in patients with chronic heart affectations [54]. The use of these beta-blockers could also reduce ACE2 activity, thus preventing the correlation of ACE2 activity and cardiac remodeling that reflects the severity of heart disease [49]. These results could be guiding and encouraging as an option in the treatment of patients with both diseases; however, a more detailed investigation is needed.

The ACE2 receptor plays a crucial role during infection and pathogenesis with *T. cruzi*, since increased levels of this receptor in the blood serum are significantly correlated with cardiac damage and remodeling. Therefore, this increase would present an ideal environment for infection by SARS CoV-2, since, as mentioned earlier in the text, this receptor is also key for the entry of the virus into cells and its consequent infection. This leads to projecting an alarming scenario for patients with Chagas disease who are in the chronic phase with both symptomatic and asymptomatic characteristics, and therefore, they are also highly susceptible to coinfections with SARS-CoV-2, which can result in an increase in mortality and morbidity.

It is important to identify the clinical characteristics of Chagas disease and COVID-19 (Table 1), take prophylactic measures towards patients at risk and contribute to the knowledge in medicine and current treatments focused on patients with CCA in order to avoid the possible complications that the association of both diseases entails. It should be taken into consideration that Chagas disease is classified as an NTD, and COVID-19 is a new disease under constant study, so it is important to contribute and increase reliable and recent data for both diseases.

## 4. Potential Risk of Heart Damage in Population with Chagas Disease or Other Comorbidities in the COVID-19 Pandemic

A high percentage of the Mexican population is at risk of developing health complications, since the COVID-19 pandemic has proven to be nondiscriminatory and to affect practically all sectors of the population in this country, with 1.4 million cases in 2020 [12]. In addition to this situation, the following factors can be added: (i) Mexico is the country with the highest rate of obesity in people over 15 years of age of the 37 member countries of the Organization for Economic Cooperation and Development (OECD), with 73% of the population being overweight and 34% of them suffering from morbid obesity, reducing life expectancy by 4.2 years [58]; (ii) a growing elderly population with approximately 8.2% of the population over 65 years of age [59]; (iii) an annual 24% of deaths are caused by cardiovascular diseases, which may be related to the fact that 17% of the Mexican population smokes and 22.8% suffers from hypertension [60] (iv) and an estimated 30 million people are at risk of developing Chagas disease [9]. All these factors make the Mexican population highly susceptible to increasing the mortality rate in COVID-19 from 2.2% worldwide to 8.5% [5], mainly among the most vulnerable population, adults over 41 years old with the aforementioned comorbidities or with harmful habits to the health such as smoking or sedentary lifestyle [61]; also, in our country there are regions with extreme poverty [62] or no access to healthcare, even more so in the context of an insufficient health system, where it is complicated to provide the population with high quality health care.

In this context, it is difficult to estimate the total number of people at risk of contracting both diseases; however, it is not difficult to imagine that Mexican society is very susceptible, due to all these conditions that propitiate an ideal scenario for the increase of mortality and morbidity in the population, which translates into a direct impact towards the most vulnerable sectors and high economic costs in health for the government.

## 5. Influence of COVID-19 Treatment in Patients with Chagas Disease

Currently, there are only two drugs approved for the treatment of Chagas disease, Nifurtimox and Benznidazole, which are based on the indirect inhibition of DNA, RNA and protein synthesis due to oxidative damage by the releasing of reactive oxygen species, resulting from the reaction of oxygen with unstable nitroanion metabolites, leading to the inactivation of enzymes and destruction of the parasite [63]. Although the mechanism of both drugs is not completely elucidated, it should be considered that both are highly hepatotoxic and genotoxic; additionally, these drugs are only highly effective in patients during the acute phase; as the disease progresses towards chronicity, their effectiveness decreases drastically [64,65]. Due to the damage derived from treatments with these drugs, it is necessary to consider the possible interactions that may exist with current COVID-19 treatments and the potential health effects in infected patients.

Unfortunately, the antiparasitic drugs are not useful in the chronic phase, and their cure rate is highly variable, which usually does not reach 10%, and also produces a significant number of side effects. In Mexico, the medical care of patients with CCC has two aspects, those patients who are admitted to a cardiology referral center in conditions that are controllable through the outpatient consultation and remain in this service, with a hospital internment; and those with decompensated and disabling heart failure, who require intensive care to stabilize them, and in whom the follow-up after the hospital stay must be closer [66]. Some patients with CCC are treated with palliative medications to help cope with the symptoms of cardiac involvement, or they are candidates for the implantation of cardioverter defibrillators for the primary and secondary prevention of sudden death in this vulnerable population; however, its effectiveness is controversial due to the possible triggering of ventricular arrhythmic storms [67]. Therefore, the consideration that must be taken is also towards the possible interactions that may exist with current treatments against SARS-CoV-2 and palliative treatments or procedures for patients with chronic phase Chagas disease; such treatments are not exactly trypanocidal drugs.

The management of both infections is of vital care and should be further studied in depth, especially for COVID-19, through constant updates of data, effects caused by both treatments and continuous contributions from experts in the field; therefore, we suggested taking into account and following the guidelines made by Zaidel E.J. et al. [37], who explained in detail the aspects of etiological treatment and provided recommendations for Chagas disease in the context of COVID-19 coinfection. They highlight the importance of postponing the start of treatment against Chagas disease in indeterminate and chronic stages, especially if there is a COVID-19 coinfection that presents symptoms or is under treatment with immunosuppressive drugs; only if the patient is asymptomatic positive and currently has initiated antiparasitic treatment for Chagas disease it is recommended to continue with it. In the acute stage of Chagas disease, the situation is more complex, since this is the ideal period for treatment and elimination of the parasite, either with Nifurtimox or Benznidazole, and it is recommended to start treatment even if there is a COVID-19 coinfection, with or without symptoms. As for the reactivation status in patients with Chagas disease, it is recommended to start treatment, even if there is a coinfection with or without symptoms, since it is also a short and critical period for the elimination of the parasite. It is important to consider that priority is given to the treatment of COVID-19, mainly because of the intensity and damage that it can cause in some patients in a very short time, whether cardiac, pulmonary or generalized; unlike Chagas disease, which, with the exception of the acute phase, usually presents very prolonged periods of affectation, reaching chronicity [37].

As we have already mentioned, cardiac muscle damage is a common feature of both diseases, which can be better understood by examining these alterations and their effects on the immune system, such as TNF-alpha and IL-6 levels, as markers of cardiac inflammation caused by both diseases and, also, the serum levels of atrial natriuretic peptides, and the ACE2 receptor levels, which is the main entry pathway for SARS CoV-2 [7,68]; the observation of these pathways in common is fundamental, since there are, for example, medications to treat CCC that inhibit or block the ACE2 receptor [37], and although there is no experimental scientific evidence of direct interactions, they should be taken into account in patients with these comorbidities. Other drugs to take into consideration in CCA are those with an immunosuppressive effect, mainly due to the immunological environment that, in infected patients, could lead to a reactivation of parasitemia, a better environment for disease and a higher probability of cardiac damage, such as: azithromycin, chloroquine, hydroxychloroquine, ribavirin, ivermectin, cyclosporin A, sirolimus, colchicine, ritonavir and lopinavir [28,43,69]. Although interactions may or may not exist, special emphasis is recommended to monitor these treatments in order not to adversely affect the patient’s health.

## 6. Recommendations for the Prevention of Chagas Disease and COVID-19

Several factors contribute to the development of Chagas disease in Latin America and Mexico; some of them are globalization, old age, poverty, limited access to healthcare, lack of knowledge of the disease and basic hygiene practices, congenital/oral transmission and underreporting [70]. However, there are prophylactic measures that are available to the whole population that could prevent the spread of this disease; these include: vector control measures, such as systematic pesticide applications in domestic and peri-domestic areas [71]; constant improvements in houses made of natural materials, such as mud, adobe or straw, which serve as natural reservoirs for triatomines; increased political and community commitment to the development and implementation of programs for the prevention, control and eradication of vectors, as well as opportune reports of infection and transmissibility among endemic or high incidence communities [43]; the strengthening of health systems for the opportune detection of Chagas disease in acute and chronic patients, as well as access to treatment if necessary; screening for Chagas disease in blood banks and in those patients who undergo organ transplants to avoid transmission through these routes, in accordance with the mandatory NOM-253-SSA1-2012 regulation [9]; universal serological screening in women of childbearing age and antiparasitic treatment before pregnancy in infected women, since it has proven effective in decreasing congenital transmission for future pregnancies [72]; health promotion campaigns to raise awareness of the disease in the general population and, particularly, in regions of high incidence and recommendations on the safe handling of food and proper hygiene before ingestion to avoid oral transmission, which, although uncommon, exists a latent risk of being infected through the feces of the triatomines [15].

In terms of prevention for COVID-19, the information is recent and constantly updated; however, there are simple and effective measures that are available to everyone that can help to curb the massive spread and, therefore, the pandemic. Some of these are: hand washing for 20 s or use of hand sanitizer before putting on the mask and constantly after touching objects and surfaces of common use; the mandatory use of a mask in people older than two years—adjust the mask correctly ensuring to cover the nose, mouth and chin; stay at least two meters away from other people who do not live with you, in addition to using masks all the time; avoid crowds and contact with people presenting symptoms; stay at home whenever possible and go out only if it is strictly necessary, always wearing a mask; avoid poorly ventilated spaces such as bars, gyms or restaurants, as these increase the possibility of catching the virus through the air; avoid shaking hands or hugs; cover coughs or sneezes with a tissue or inside the elbow; after throwing away the tissue, wash hands immediately for 20 s or disinfect with sanitizer; constantly clean and disinfect commonly used surfaces with soap and then with disinfectants or sanitizers, always being careful not to mix them to avoid intoxication or accidents; monitor your health constantly—be alert to possible symptoms, and in case of them, go immediately to a doctor; check signs such as oxygenation and temperature if it is possible and, finally, try to maintain a healthy nutritional, physical and mental lifestyle. Although many of these measures have proven to be highly effective and simple to implement, some sectors of the population around the world have not followed them, which has contributed to the rapid spread of the virus, increasing the mortality and morbidity rates in Mexico and the world [73,74,75]. Additionally, there is great evidence in the literature that vitamin D is beneficial for the proper functioning of the immune system by fighting pathogens and preventing autoimmune diseases. At this time of the pandemic, until July 2020, there were 21 ongoing Vitamin D clinical trials for COVID-19 with detailed information showing that it is highly favorable, while there is currently no specific therapy for COVID-19 available [76] for the entire Mexican population.

Taking the corresponding measures for these diseases has a fairly good cost–benefit ratio, since they can avoid many complications, as they can be carried out by practically the entire population, and although they cannot guarantee not contracting the disease, they are very effective reducing the risk of acquiring them, either in people who suffer from any of them or none at all. Therefore, it is important to know and be aware of all the actions available and the impact they can have to prevent or reduce both diseases in oneself and in society in general.

## 7. Conclusions and Perspectives

Historically, Chagas disease has been a neglected tropical disease, and the lack of a characteristic clinical symptomatology does not allow its identification, and therefore, its registration in the acute phase is underestimated. In addition, in the chronic phase, in which patients can develop the affectations known as megas, including chronic Chagasic cardiopathy, it is of vital importance for health agencies in each country and in Mexico to carry out identification and registration in order to understand the situation and to establish appropriate measures to reduce the risks associated with this disease and, especially, to determine how to act in the case of patients with chronic Chagas disease and who may have other comorbidities or coinfections, such as with SARS-CoV-2.

On the other hand, the current COVID-19 pandemic has brought serious consequences for the general population; however, there are no studies on how this pandemic has affected patients with Chagas disease and, moreover, with cardiac involvement that can range from moderate to severe, with the presence of the SARS-CoV-2 virus, where these conditions can be a major trigger to cause death. There are also few studies on the impact that the current treatment has on the COVID-19 disease and how it might impact patients with Chagas disease. Therefore, it is important that several research groups worldwide could evaluate these patients with coinfections and, thus, have conclusive data that allow an adequate management of patients with Chagas disease, as, from now on, SARS-CoV-2 or other viruses will be pathogens with which the entire world population will coexist.

## Figures and Tables

**Figure 1 biology-10-00411-f001:**
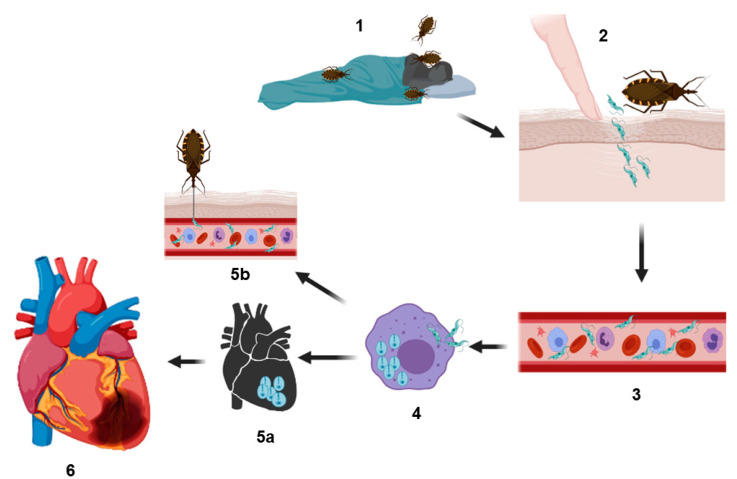
Representation of the biological cycle of *Trypanosoma cruzi*. Infection is generated when hematophagous bugs reach a mammalian host to feed (**1**), infected bug feces (containing the parasite in their form as metacyclic trypomastigotes) are released on the skin (near the bite wound) or in the mucosa of the host. Through micro-wounds caused by scratching excessive or by mucosa route, the parasite reaches the bloodstream (**2** and **3**) and invades different types of nucleated cells (**4**) (phagocytic and nonphagocytic). The parasitophorous vacuole is formed inside the infected cells to eliminate the parasite; however, *T. cruzi* trypomastigotes escape this vacuole and differentiate in amastigotes, which are the replicative forms of the parasite in the mammalian host. After several rounds of replication in the cytosol, the amastigotes differentiate into trypomastigotes, and then, the infected cells burst, and the parasites are subsequently released into the bloodstream where they can disseminate to several tissues, mainly the heart, colonizing the cardiac cells and forming the so-called amastigote nests (**5a**), or they can reinvade the blood cells. Finally, the circulating forms can be taken up by a new triatomine vector during a blood meal, and this vector can infect to another healthy host (**5b**). In other hand, cardiac affectations can be observed in the acute phase or in the chronic phase; these affectations can be in different degrees of severity (**6**) (see text) culminating in a heart attack, which can cause the death of the host.

**Figure 2 biology-10-00411-f002:**
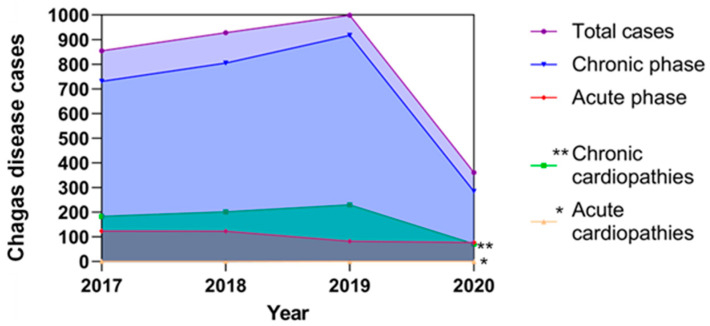
Reported cases of Chagas disease in Mexico and their possible correlations with cardiopathies from the period 2017–2020.

**Table 1 biology-10-00411-t001:** Clinical features on cardiac affectations due to COVID-19 and Chagas disease.

COVID-19	Chagas Disease
Rhythm abnormalities: supraventricular and ventricular tachyarrhythmias, bradyarrhythmias and chest pain or chest tightness on exertion [47].	Rhythm abnormalities: bradyarrhythmias and conduction system abnormalities and atrial/ventricular tachyarrhythmias [55].
Electrocardiogram abnormalities: QT prolongation, pseudo infarct pattern and premature ventricular complexes. Development of cardiac arrest and variations in the myocardial enzyme level [30,47,56].	Primary ST- and T-wave abnormalities, pathological Q waves or electric inactive areas [55]. Segmental/global wall motion abnormalities, dilated cardiomyopathy and function mitral/tricuspid regurgitation [21,55].
Acute coronary syndrome, venous thromboembolism and elevated levels of troponin T (>0.022 ng/mL) [57].	Aneurysms: left ventricular apical, other left ventricular segments (inferior and inferolateral walls) and uncommonly right ventricular [55].
Capillary endothelial cells dysfunction and induced micro-circulation disorder [34].	Thromboembolic events: ischemic attack or stroke, pulmonary or systemic emboli [22].
Pericardial inflammation, microvascular ischemia (through pericytes) and myocardial edema/scar (nonischemic) [47].	Microvascular abnormalities: precordial/retrosternal chest pain without evidence of epicardial coronary artery disease [22].
Prolonged myocardial inflammation and disseminated intravascular coagulation [31].	Myocardial scar/interstitial fibrosis and pericarditis [18,22].
Heart failure, including raised jugular venous pressure, peripheral edema and right upper quadrant pain [38]. Cardiac dysfunction and normal/acute injury [44].	Myocardial micronecrosis [18].
Myocarditis and fulminant myocarditis with presence of febrile and low pulse pressure, cold or mottled extremities, and sinus tachycardia [47]. Proinflammatory cytokines production (IL-6 and TNF-alpha) and high levels of creatinine kinase (>200 U/L) [34,38].	Myocarditis associated with myocytolysis, myofiber hypertrophy and cardiac remodeling, resulting in a fully enlarged heart [18,21].

## Data Availability

The data presented in this study are available upon request from the corresponding authors.

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
