# Peer review of "Risk of COVID-19 in Chagas Disease Patients: What Happen with Cardiac Affectations?"

_biology, 2021, doi:10.3390/biology10050411_

Round 1
Reviewer 1 Report
After a detailed evaluation, considering the biological importance of the subject, the manuscript should be accept after small corrections:
- English must be corrected by a native professional, especially in the first sections (up to page 5);
- The abreviation "T. cruzi" follows the international rules for citation of biological species, and doesn´t need to be include inside parenthesis (line 44, introduction). Please delete it;
- Lines 61-62: "...which makes the patient more susceptible to infection by the COVID-19" COVID-19 should be replaced by Sars-CoV-2; Are there studies about it? I´m sure if something like that could be said, without the randomized experiment be performed;
- Who is the main vector of Chagas disease in Mexico? (line 72). In the sentence, the authors described as triatomines... In generic context, triatomine is the only vector, not main...
- Line 81: "this pandemic" replace by "covid-19 pandemia"
- Line 94: Replace "lineage" by "DTU" including the concept of DTUs...
- The parasite cycle shows misinterpretation. "In the acute phase of heart Chagas disease of mammals, the process initiates with the inoculation of the parasite, where it will begin to replicate near the inoculation site (lines 143-144)." Metacyclic trypomastigotes reached bloodstream (inoculation suggest the direct inoculation), and trypomastigotes are non-replicative form of the parasite. In the figure 1 legend, "After several rounds of replication, the infected cells burst and the parasites are released into the bloodstream (lines 165-166)" the authors did not mention the intracellular differentiation amastigotes to trypomastigotes, and subsequent release of these latter forms in the bloodstream. In summary, the part of T. cruzi life cycle must be revised carefully.
- "the most common T. cruzi strain found in Mexico has a marked tropism towards cardiac muscle (lines 174-175)" Please include the strain name.
- Lines 180-181: Avoid the term "pseudo-cysts"... It is well-known that trypanosomatids do not present true cystic forms as giardia for example, and the term pseudo-cysts is used for trichomonas for a particular case far from what is observed in T. cruzi infection. Due to the fibrosis, extracellular matrix may isolate parasites from "external environment". Please correct it.
- Line 247: Replace "chagas" by "Chagas disease";
- I also suggest a inclusion of a small discussion about the necessity of the compulsory notification of chronic cases of Chagas disease, despite all difficulties involved.
Author Response
Thank you very much for the feedback to our manuscript. We are sure that all comments were of great value to enhance our work quality. Next, you will find our responses to your points and the indications of the changes made to the manuscript. Please note that the Track Change function in the Word file of the manuscript was employed; thus, lines mentioned in the responses correspond to the version that shows the made changes.
Reference list have changed (and numbers in the text citations) due to text deletions/additions and the addition of new references.
We hope that this revised version of the work is now suitable for publication.
Best regards.
- English must be corrected by a native professional, especially in the first sections (up to page 5);
R: We appreciate your suggestion to revise the English, unfortunately the editor only gave us 5 days to do the revision and we could not send it to a native speaker, we revised the English thoroughly. However, if you consider it necessary, we will be sent to the English edition by the journal asking the editor for an adequate time for the revision.
- The abreviation "T. cruzi" follows the international rules for citation of biological species, and doesn´t need to be include inside parenthesis (line 44, introduction). Please delete it;
R: We appreciate your clarification; we have deleted the abbreviation.
- Lines 61-62: "...which makes the patient more susceptible to infection by the COVID-19" COVID-19 should be replaced by Sars-CoV-2; Are there studies about it? I´m sure if something like that could be said, without the randomized experiment be performed;
R: According to your indication we have made the change from COVID-19 to SARS-CoV-2, since the infectious agent is the virus and the disease that produces it is called COVID-19.
Regarding what makes the patient with Chagas disease more susceptible to the SARS-CoV-2 virus. We have added a section in point 3.0 Cardiac implications, patients with COVID-19 and Chagas disease. This new section is about the increase of the ACE2 receptor in patients with Chagas disease and its possible correlation with susceptibility to virus infection. Lines 354-395 in the new version.
- Who is the main vector of Chagas disease in Mexico? (line 72). In the sentence, the authors described as triatomines... In generic context, triatomine is the only vector, not main...
R: We agree with your comment about there is only one vector; we have corrected the sentence.
- Line 81: "this pandemic" replace by "covid-19 pandemia"
R: According to your indication we have made the change from “this pandemic” to “COVID-19 pandemia”
- Line 94: Replace "lineage" by "DTU" including the concept of DTUs...
R: We agree with your comment, the term lineage is not currently used and the proper one is DTU. We have corrected the term and added a small definition about what DTUs are. Lines 94-95: Discrete Typing Unit, DTU (molecular genotyping that allows to identify particular characteristics of the isolated trypanosomes)
- The parasite cycle shows misinterpretation. "In the acute phase of heart Chagas disease of mammals, the process initiates with the inoculation of the parasite, where it will begin to replicate near the inoculation site (lines 143-144)." Metacyclic trypomastigotes reached bloodstream (inoculation suggest the direct inoculation), and trypomastigotes are non-replicative form of the parasite. In the figure 1 legend, "After several rounds of replication, the infected cells burst and the parasites are released into the bloodstream (lines 165-166)" the authors did not mention the intracellular differentiation amastigotes to trypomastigotes, and subsequent release of these latter forms in the bloodstream. In summary, the part of T. cruzi life cycle must be revised carefully.
R: We appreciate your comments, we have reviewed carefully theT. cruzi life cycle and have edited it. You will observe the changes marked in the new text.
- "the most common T. cruzi strain found in Mexico has a marked tropism towards cardiac muscle (lines 174-175)" Please include the strain name.
R: We appreciate your comment. In this sentence, we are not really referring to a particular strain but to the most common DTU present in Mexico, which has a marked tropism for the heart muscle. We have corrected the sentence to amend the mistake (line 198 in the new version).
- Lines 180-181: Avoid the term "pseudo-cysts"... It is well-known that trypanosomatids do not present true cystic forms as giardia for example, and the term pseudo-cysts is used for trichomonas for a particular case far from what is observed in T. cruzi infection. Due to the fibrosis, extracellular matrix may isolate parasites from "external environment". Please correct it.
R: Thanks for your comment, you are totally right, it is really a mistake to use the term pseudocyst, since the cyst is a form of resistance of some parasites and does not refer to the isolation of T. cruzi amastigotes in cardiac cells. The proper term is amastigote nests. So, we have corrected it in the text (lines 266-267 in the new version).
- Line 247: Replace "chagas" by "Chagas disease";
R: According to your recommendation we have changed “chagas” by “Chagas disease”.
- I also suggest a inclusion of a small discussion about the necessity of the compulsory notification of chronic cases of Chagas disease, despite all difficulties involved.
R: We appreciate your suggestion, which is an important issue and must be heard by the authorities in each country where this disease occurs, so we have written a recommendation in section 4.0 conclusions and perspectives about it. Lines 565-573 in the new version.
Reviewer 2 Report
I commend the authors for such extensive work and for taking into consideration most of the aspects of such complex disease.
I would foresee that concomitance with the Covid19 pandemic may offer a leverage for more research and possibly for using biotechnologies that are showing to be successful with it; on the other hand Chagas disease would probably remain for a longer time and the overlap with Covid19 has different implication in the short term for topical treatments or prevention strategies as stated by the Authors so I would consider clarifications or comments about the need to use Covid19 developed treatments or prevention strategies to improve future management of Chagas.
In addition I would suggest to exploit or discuss more cardiac implications in Chagas beyond the heart muscle dilatation, for example arrhythmias; i this would offer the mean to discuss also innovative therapies to target those like cardiac ablation or pre-empitive use of ICDs
Author Response
Thank you very much for the feedback to our manuscript. We are sure that all comments were of great value to enhance our work quality. Next, you will find our responses to your points and the indications of the changes made to the manuscript. Please note that the Track Change function in the Word file of the manuscript was employed; thus, lines mentioned in the responses correspond to the version that shows the made changes.
Reference list have changed (and numbers in the text citations) due to text deletions/additions and the addition of new references.
We hope that this revised version of the work is now suitable for publication.
Best regards.
I would foresee that concomitance with the Covid19 pandemic may offer a leverage for more research and possibly for using biotechnologies that are showing to be successful with it; on the other hand Chagas disease would probably remain for a longer time and the overlap with Covid19 has different implication in the short term for topical treatments or prevention strategies as stated by the Authors so I would consider clarifications or comments about the need to use Covid19 developed treatments or prevention strategies to improve future management of Chagas.
In addition I would suggest to exploit or discuss more cardiac implications in Chagas beyond the heart muscle dilatation, for example arrhythmias; i this would offer the mean to discuss also innovative therapies to target those like cardiac ablation or pre-empitive use of ICDs.
R: We sincerely appreciate your suggestion. You are totally right, and we agree with your comments; however, we want to focus on the cardiac involvement caused by the two diseases and which are very similar (table 1), so they can be confused at the time of diagnosis. We think your suggestion is very good and it would require writing a complete paper focused on all the cardiac affectations that can occur in the chagasic patient and what could be the opportune treatment for each one of these affectations.
Reviewer 3 Report
The review article entitled “Risk of COVID-19 in Chagas Disease Patients: What happen with cardiac affectations?” by Alejandro Diaz-Hernandez et.al, discusses the effect of COVID 19 and chagas diseases on cardiac implications and the role of ACE2 in SARS-COV2 entry. The review article summarizes the pathology of acute and chronic Chagas disease in detail. The authors discuss the potential risk of heart damage in Chagas patients due to COVID19 infection and the effects of covid19 treatment related to inflammation. However, the article does not discuss the role and the effect of ACE2 levels in Chagas patients and how it may influence the outcome of COVID19 disease in these patients. Given that the plasma levels of ACE2 correlates to cardiac severity in Chagas patients, these information in more detail is essential. Otherwise, the review article just summarizes the information which is already available.
Author Response
Thank you very much for the feedback to our manuscript. We are sure that all comments were of great value to enhance our work quality. Next, you will find our responses to your points and the indications of the changes made to the manuscript. Please note that the Track Change function in the Word file of the manuscript was employed; thus, lines mentioned in the responses correspond to the version that shows the made changes.
Reference list have changed (and numbers in the text citations) due to text deletions/additions and the addition of new references.
We hope that this revised version of the work is now suitable for publication.
Best regards.
The review article entitled “Risk of COVID-19 in Chagas Disease Patients: What happen with cardiac affectations?” by Alejandro Diaz-Hernandez et.al, discusses the effect of COVID 19 and chagas diseases on cardiac implications and the role of ACE2 in SARS-COV2 entry. The review article summarizes the pathology of acute and chronic Chagas disease in detail. The authors discuss the potential risk of heart damage in Chagas patients due to COVID19 infection and the effects of covid19 treatment related to inflammation. However, the article does not discuss the role and the effect of ACE2 levels in Chagas patients and how it may influence the outcome of COVID19 disease in these patients. Given that the plasma levels of ACE2 correlates to cardiac severity in Chagas patients, these information in more detail is essential. Otherwise, the review article just summarizes the information which is already available.
R: We appreciate your comment, in the text we had mentioned the importance of the ACE2 receptor for the entry of the SARS-CoV-2 virus; however, we had not established the correlation that could exist between this receptor in patients with Chagas disease and its possible susceptibility to get infected with the virus. Therefore, in the new version of the text, we have described the importance of this ACE2 receptor and the increased susceptibility that patients with Chagas disease may have to acquiring the SARS-CoV-2 virus. Lines 354-395 in the new version.